# An Integrated Spatio-Temporal Features Analysis Approach for Ocean Turbulence Using an Autonomous Vertical Profiler

**Xiuyan Liu** [1] **, Dalei Song** [2] **, Hua Yang** [3] **, Xiaofeng Wang** [1] **and Yunli Nie** [4,*]

1 School of Information and Control Engineering, Qingdao University of Technology, Qingdao 266525, China; liuxiuyan@qut.edu.cn (X.L.); wxf846643042@163.com (X.W.)
2 College of Engineering, Ocean University of China, Qingdao 266100, China; songdalei@ouc.edu.cn
3 College of Information Science and Engineering, Ocean University of China, Qingdao 266100, China; hyang@ouc.edu.cn
4 College of Ocean Science and Engineering, Shandong University of Science and Technology, Qingdao 266590, China
* Correspondence: nieyunli@sdust.edu.cn

**Abstract:** Turbulent energy cascade and intermittency are very important characteristics in the turbulent energy evolution process. However, understanding the temporal–spatial features of kinetic energy transfer and quantifying the correlations between different scales of turbulent energy remains an outstanding challenge. To deeply understand the spatial–temporal features in the energy transfer process, an integrated features identification and extraction method is proposed to quantitatively investigate the correlations using the ocean shear turbulence measured by an autonomous vertical reciprocating profiler (AVRP). The proposed integrated method mainly contains two parallel features analysis modules: first, temporal multiscale features structures of the nonlinear and nonstationary turbulent cascade are identified by Variational Mode Decomposition (VMD); then, the ocean microstructure shear fluctuation data are decomposed into a series of intrinsic mode functions (IMFs), which are characterized by different time scales and frequency bandwidths. The local features of energy transfer are identified when the local intermittency peaks overlap and the phase-synchronization case occurs between two neighboring scales; second, the spatial statistical characteristics of the turbulent energy dissipation are quantitatively studied. The cumulative probability distribution functions (CPDFs) of kinetic energy dissipation are approximated well by a normal distribution, indicating that the turbulent dissipation process exhibits a robust spatial scaling correlation and a few intense dissipation locations dominate the integrated process. Finally, the proposed integrated method is evaluated through experiments using an autonomous vertical reciprocating profiler deployed in the South China Sea. Preliminary experimental results show that the proposed novel method is useful to improve our understanding of turbulent energy transfer and the evolution process in the ocean dynamic systems.

**Keywords:** ocean turbulence; energy cascade; dissipation rates; spatio-temporal features; VMD

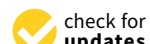



## 1. Introduction

In recent years, the features of turbulent energy cascade and dissipation rates in turbulent flows have become one of the hot research fields in ocean dynamics. Therefore, oceanographic communities have paid considerable attention to study the mechanism of energy cascade and eddy diffusivity in the spatial distribution [1]. The essence of the turbulent kinetic energy (TKE) evolution process is the conveyance of energy from the largest scales to the smallest scales, where it is dissipated through the action of kinematic molecular viscosity [2]. The phenomenon of nonlinear energy cascades and intermittency will happen in the complicated energy evolution and mixing process. This complicated energy evolution process in turbulent flows is described as follows. First, the largest scale vortexes in the turbulent energy injected area will be broken to some mesoscale vortexes,

and these largest vortexes will convey energy to the smaller-scale vortexes through inter-vortexes transformation in this energy transfer process. Second, these mesoscale vortexes in the turbulent energy transfer area will be broken into some microscale vortices; at the same time, the energies that these mesoscale vortexes contained will be transferred into the microscale scales. Finally, the microscale vortexes in the energy-dissipating region will be dissipated continuously due to the viscosity action. In the integrated broken process, turbulent energy is transferred from larger eddies to the relative smaller scales before dissipating in the microscale, and the TKE is transformed into the heat energy of the fluid, which is termed as dissipation. Due to the effect of the boundary action, disturbance, and velocity gradient, new eddies are constantly generated [3,4]. A literature review has shown that this energy cascade process is not steady but intermittent and nonstationary. When the turbulent energy is transferred into smaller-scale vortexes, some strong intermittent bursts of activity make large contributions to the intermittency of the turbulent flows [5,6]. Research on the characteristics of energy cascade and the intermittency play an essential role in understanding the energy evolution mechanism of ocean turbulence and make great efforts toward a complete theory of energy cascade. However, it is an outstanding challenge to capture and understand the organization of a multiscale energy transfer process.

Currently, the common methods to study turbulence properties are multifractals, scale-dependent, multiplicity of scaling exponents, the structure functions, coherent structures, and statistics of discontinuities. Onorato et al. [5] developed a new method based on an extended self-similarity and the probability density function of the wavelet coefficients to study the intermittency properties of turbulent flows. Xu et al. [7] performed a 2D continuous wavelet transform (WT) to study the scale-dependency of turbulence intermittency in a directly simulated turbulent channel flow and describe the process of the intermittent nature of the turbulence energy dissipation. Foucher and Ravier [8] proposed a complementary technique based on the empirical mode decomposition (EMD) to determine turbulence properties. Arbic et al. [9] examined the spectra fluxes of turbulent kinetic energy in the frequency domain to quantify the nonlinear cascades. However, their conclusions make it difficult to understand whether the forward temporal cascades seen in some regions in altimeter data represent physics that is missing in the models studied here or merely sampling artifacts. The next year, Arbic et al. [10] explored the impacts of horizontal eddy viscosity and horizontal grid resolution on geostrophic turbulence. Zhdankin et al. [11] used a numerical simulation method to obtain the magnetohydrodynamic (MHD) turbulence data to further study their temporal features of intermittency and energy dissipation. Camporeale et al. [12] employed a space-filtering technique to quantify the inhomogeneity of the turbulence and the spectral energy flux. Wang et al. [13] proposed an extended variational mode decomposition (VMD) method to discuss the multiscale characteristics of multi-component turbulence. Vásconez et al. [14] studied the global and local properties of energy exchange and cascade in plasmas turbulence using the direct numerical simulations (DNS) model. It's worth noting that most of these aforementioned methods utilize the simulated turbulence data rather than the data observed in the actual deep-sea fields. Although some efforts have made to reveal the mechanism of turbulent energy transfer, there is no mechanistic framework that helps us understand how turbulent flows cascade and transfer.

To quantitatively characterize the process of the turbulent dissipated energy among a wide range of turbulent scales and further examine the different physical processes occurring on these different time scales, an integrated spatio-temporal synchronous quantitative analysis method is innovatively proposed to exhibit the properties in energy transfer and the cascade process. Observed turbulence shear data using an autonomous VRP instrument are utilized to reveal the mechanism of kinetic energy cascade and the interactions of multiscale energy dissipation in an oceanic system.

The rest of this article is arranged as follows. The actual field experiment of an autonomous VRP deployed on the mooring system is briefly described in Section 2. In Section 3, an integrated method combined with the VMD decomposition, the multiscale

wavelet transform, and a local intermittency measure is proposed to capture the spatio-temporal intermittency features of energy transfer in turbulent flows. The preliminary results are presented in Section 4, including the time-frequency and wavenumber spectrum features, the temporal features of energy cascade and intermittency, and the spatial features of dissipation rates. Subsequently, some conclusions and future work are finally given in Section 5.

## 2. Experiments

The spatial–temporal properties of turbulent energy cascade were analyzed using the real experimental dataset collected by an autonomous vertical reciprocating profiler (AVRP), which was designed independently by the Ocean University of China, and the detailed mechanical structure and working principles of this AVRP platform have been given in the literature [15]. The deployed moored system (Figure 1) comprises several buoyancy elements, a 38 kHz acoustic doppler current profiler (ADCP), two groups of Sea-Bird Electronics CTD sensors (SBE37SM-RS232, recorded the information about conductivity, temperature and depth), an AVRP instrument, a Doppler current meter (RCM11), two parallel acoustic releases (ARs), and an anchor block [15]. The CTD sensor mainly records the water depth, temperature, and conductivity in the vertical direction. Two groups of buoyancy elements are respectively installed at 600 m and 1500 m to determine the depth limitation of the profiler instrument. The RCM instrument is used to measure the flow velocity and flow direction. The acoustic releases are attached to receive the command to control the moored system. The anchor block provides gravities to make a balance for the moored system. Two orthogonal shear probes (PNS07) and a vibration sensor (MTI-300) are installed on the vertical VRP profiler, where shear probes are used to measure the shear velocity fluctuations, and an MTI sensor is utilized to record three orthogonal accelerations (horizontal acceleration ($Ax$), the lateral acceleration ($Ay$), and the vertical acceleration ($Az$)) and three attitudes of profiler (Heading ($\psi$), Pitch ($\varphi$), and Roll ($\theta$)). The calibration procedure between two orthogonal shear data and three accelerations are elaborated by Fer and Paskyabi [16]. The autonomous AVRP instrument was deployed at 118°09.83′ E 20°55.49′ N (Figure 2) in the South China Sea (SCS) to measure the turbulent microstructure shear, and the position of the AVRP (rectangle) is shown. The AVRP platform rises and sinks repeatedly along a mooring cable to measure the turbulence shear between 600 and 1500 m in deep sea, and it is a relatively low-noise environment. Therefore, the measured turbulence data are a little affected by the wave and tidal current. In this article, we mainly concentrate on the vortexes induced by the unsteady separation of turbulent flows. The formation and shedding of the turbulent vortex is an alternate process, in which the turbulent energy will transfer and dissipate.

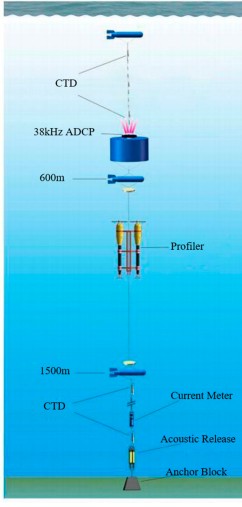

**Figure 1.** The sketch of the mooring system of a vertical profiler.

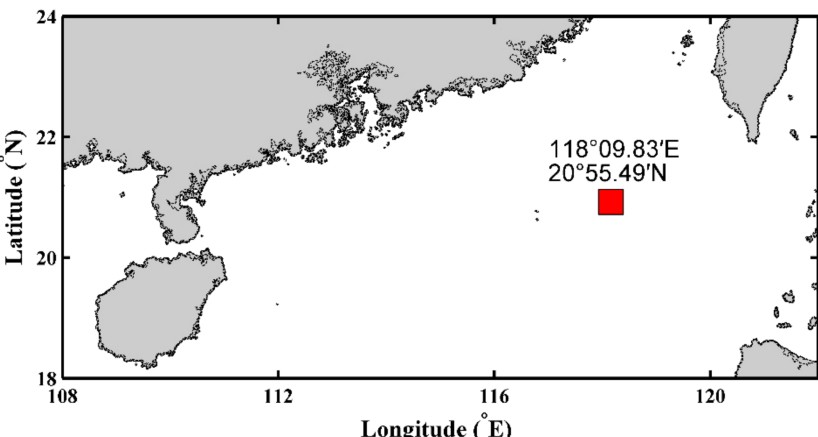

**Figure 2.** Deployment site, the location of the vertical profiler in SCS (red square).

## 3. Methodologies

### 3.1. The VMD Decomposition

Currently, the empirical mode decomposition (EMD) method [17,18] has been widely applied to the physical oceanography fields, such as experimental turbulence records. It is especially effective to decompose the nonlinear and nonstationary time series. The essence of the EMD method is that it decomposes the original signals into several IMF modal components with different scales and frequencies, and these different local features on different time scales can identify the structures of the turbulent cascade and intermittency through the dissipative power spectrum. However, EMD decomposition has several disadvantages, such as mode mixing and an endpoint effect. To reduce the effect on the mode mixing problem, Dragomiretskiy et al. [19] proposed a new non-recursive variational mode decomposition (VMD) method, which decomposed the original signal into K sparse amplitude–frequency modulation components by solving the optimal solution of the variational problem [20]. The variational model can be expressed as follows:

$$
\begin{cases}
argmin\left\{ \sum_{k=1}^{K} \|\partial_t\left[\left(\delta(t) + \frac{j}{\pi t}\right) \times u_k(t)\right]e^{-i\omega_k t}\|_2^2 \right\} \\
s.t. \sum_{k=1}^{K} u_k(t) = f(t)
\end{cases}
\tag{1}
$$

where $u_k(t) = A_k(t)\cos(\varphi_k(t))$ denotes the intrinsic mode functions (IMF) components. $\{\omega_k\} = \{\omega_1, \omega_2 \ldots .\omega_K\}$ represents the frequency centers of each $u_k(t)$, and K represents the total number of IMF components. In order to solve the optimal solves in Equation (1), two parameters of an extended Lagrange operator $\lambda(t)$ and a second penalty factor $\alpha$ are developed to transform the constrained variational model into a non-constrained variational problem, and the Lagrange operator $L(u_k, \omega_k, \lambda_k)$ is depicted as follows:

$$
L(u_k, \omega_k, \lambda_k) = \alpha \sum_{k=1}^{K} \|\partial_t\left[\left(\delta(t) + \frac{j}{\pi t}\right) \times u_k(t)\right]e^{-i\omega_k t}\|_2^2 + \|f(t) - \sum_{k \neq 1}^{K} u_K(t)\|_2^2 + \left(\lambda(t), f(t) - \sum_{k=1}^{K} u_k(t)\right)
\tag{2}
$$

$$
\begin{cases}
\hat{u}_k^{n+1}(\omega) = \frac{\hat{f}(\omega) - \sum_{i<k} \hat{u}_i(\omega) - \sum_{i>k} \hat{u}_i^n(\omega) + \hat{\lambda}(\omega)/2}{1 + 2\alpha(\omega - \omega_k)^2} \\
\omega_k^{n+1} = \frac{\int_0^\infty \omega|\hat{u}_k^{n+1}(\omega)|_2 d\omega}{\int_0^\infty |\hat{u}_k^{n+1}(\omega)|_2 d\omega} \\
\hat{\lambda}^{n+1}(\omega) = \hat{\lambda}^n(\omega) + \tau\left(\hat{f}(\omega) - \sum_k \hat{u}_i^{n+1}(\omega)\right)
\end{cases}
\tag{3}
$$

The method of alternating multipliers is used to update weights $\hat{u}_k$, $w_k$, and $\hat{\lambda}$ to find the optimal solution. In Equation (3), $\hat{u}_k$ and $\omega_k$ represent the modal component and center frequency, respectively. For the iteration process, the variational solution process will stop when the iteration condition $\sum_k(\|u_k^{n+1} - u_k^n\|_2^2 / \|u_k^n\|_2^2) < \varepsilon$ is satisfied, and K modal components with limited bandwidth are output.

Compared with the traditional EMD decomposition, the VMD method cannot completely remove the confusion of the modes mixing, but it has an inhibitory effect on mode mixing. Actually, the VMD method with its mathematical theoretic foundation outperforms the EMD in dealing with mode aliasing problems. Additionally, the VMD method can effectively eliminate the cumulative errors of envelope estimation caused by recursive mode decomposition. However, two parameters, the number of modal components and the penalty parameters in VMD need to be optimized by the optimization algorithm to find the optimal solution in future work. Currently, VMD is mainly utilized to obtain several IMF modes; the number of IMF modes and the penalty factor have few effects on analyzing multiscale properties in the energy transfer process.

### 3.2. Wavelet Transforms

Wavelet transform (WT), as a linear representation, splits original signals into different frequency components and time energy distribution. According to the different time and scale parameters, wavelet transform is generally divided into two kinds of decomposition formats: the discrete wavelet transformation (DWT) and the continuous wavelet transformation (CWT). As an effective signal processing technique, the WT decomposition is especially effective to decompose those nonlinear and nonstationary signals, which have different scales or singularities or short-lived transient components [21]. Wavelet function captures the time-frequency dependency using a multi-resolution and multi-level decomposition. The continuous wavelet transformation $W(a,b)$ is expressed as the inner products of the time-shifted wavelet function $\Psi_{a,b}(t)$ and the decomposed signal $f(t)$. Continuous wavelet transform is expressed as follows:

$$W_f(a,b) = \langle f(t), \Psi_{a,b}(t) \rangle = \frac{1}{\sqrt{a}} \int_{-\infty}^{+\infty} f(t)\Psi\left(\frac{t-b}{a}\right) dt \tag{4}$$

where $\Psi_{a,b}(t)$ is the mother wavelet, and the two parameters a and b respectively denote the time-scale and time-shift factor. The variable $W_f(a,b)$ is the wavelet transform coefficient that represents the similarity and correlation between the signal $f(t)$ and wavelet $\Psi(t)$. According to the characteristics of ocean turbulence, a continuous wavelet transform is performed to extract scale properties and their correlations of energy cascade and intermittency in different scales.

### 3.3. Local Measure of Intermittency

To have a quantitative identification of the temporal intermittent features of multiscale dissipated energy, a special local intermittency measure (LIM) method, as proposed by Farge et al. [22], is utilized to measure the turbulent shear flows, and the local measure of intermittency is defined as:

$$\text{LIM}_{a,b} = \frac{|W_{a,b}|^2}{\langle |W_{a,b}|^2 \rangle_b} \tag{5}$$

where $W_{a,b}$ is the wavelet transform coefficient, which is calculated by wavelet transform to extract the time $b$ and scale $a$ from each IMF mode, and the symbol $\text{LIM}_{a,b}$ denotes the ratio of the squared wavelet coefficient $|W_{a,b}|^2$ to its time average value $\langle |W_{a,b}|^2 \rangle_b$ [21]. Consequently, the equation $\text{LIM}_{a,b} = 1$ denotes that each portion of signal contains the equivalent energy spectrum; thus, the measured signals show not obvious local intermittency. Differently, the condition $\text{LIM}_{a,b} > 1$ denotes that some portions of the measured signal at these special times and scales make relatively larger contributions to the cascade and intermittency structures [6]. In conclusion, the LIM measure can make the localization and identification in those scales that contained energy above the average energy. Due to the nonlinearity and nonstationary of turbulence shear data, the features of ocean turbulent energy cascade and intermittency can be quantified using the wavelet transform and LIM approaches.

### 3.4. An Integrated Analysis Method

The features synchronous localization and identification of turbulent energy method is a combination of an integrated VMD method, a 1D continuous wavelet decomposition, and a special local measure of intermittency approach. Integrated features identification methods mainly consist of two parallel modules and the framework of a features identification and quantitation method, as shown in Figure 3.

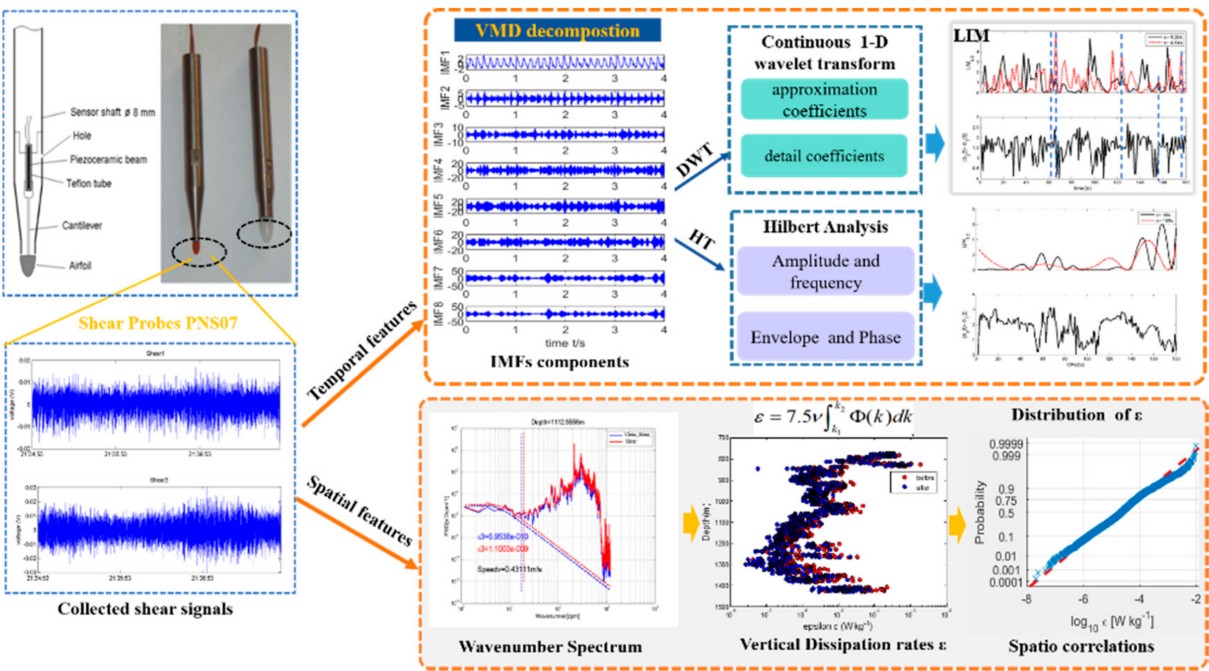

**Figure 3.** A brief framework of an identification and quantitation method.

- Temporal features localization and quantitation.

First, the measured turbulent shear fluctuations are decomposed into K IMF components associated with multiscale and multi-frequency bandwidths. Second, each IMF mode is calculated using the classical wavelet decomposition method to obtain the time and scale values of wavelet coefficients. Synchronously, the Hilbert transform (HT) is also utilized to get the phase values and unilateral frequency spectrum of IMF modes. Finally, the measure method of local intermittency is applied to identify the temporal intermittent events and scales correlations in the energy transfer process.

- Spatial features identification.

First, original horizontal shear fluctuations in the time domain are transformed into vertical shear gradients according to the empirical Taylor's frozen theorem: $\partial U/\partial z = W^{-1} (\partial U/\partial t)$ [23], where $W$ is the speed of the instrument, and the symbol $U$ denotes the shear fluctuations' velocity. Second, the shear spectrum is obtained using the fast Fourier transform (FFT) for the processed vertical gradients data $\partial U/\partial z$ with an interval of 4 s. The coherent spectrum calibration method proposed by Goodman et al. [24] is used to eliminate the interference of instrument vibration and noise, and then, the Nasmyth standard spectrum is fitted to calculate the dissipation rate ($\varepsilon$) of turbulent kinetic energy (TKE) iteratively, which is expressed as follows:

$$\varepsilon = \frac{15}{2}\nu\overline{\left(\frac{\partial U'}{\partial z}\right)^2} = \frac{15}{2}\nu\int_{k1}^{k2}\Phi(k)dk \tag{6}$$

where $\partial U'/\partial z$ is the variance of the shear fluctuations' velocity, $\nu$ ($\nu \approx 1.64 \times 10^{-6}$ m$^2$ s$^{-1}$) is the kinematic molecular viscosity, $k_1$ is the lower integration limit, and $k_2$ is the upper integration limit, which is resolved by the Kolmogorov cutoff wavenumber, where

$k_2 = \frac{1}{2\pi} \left(\frac{\varepsilon}{v^3}\right)^{1/4}$. Finally, the spatial statistical characteristics of the turbulent energy dissipation are quantitatively studied using the cumulative probability distribution functions (CPDFs) to reveal the spatial correlations in the dissipation process [25,26]. The corresponding cumulative distribution function is defined as:

$$\text{CDFs}(\varepsilon) \;=\; \Phi\left(\frac{\ln \varepsilon - \mu}{\sigma}\right) \tag{7}$$

where the symbol $\Phi$ denotes the cumulative distribution functions (CDFs) of a standard normal distribution.

## 4. Results

### 4.1. Features of Time-Frequency and Wavenumber Spectrum

The shear spectra in the time-frequency and wavenumber domain have some significant features, which are essential to corroborate the turbulence data quality and also to dominate the turbulent energy transfer. For the collected original data of shear velocity, strong fluctuations at small scales are visible (Figure 4). Naturally, when the measured shear time series is decomposed by the VMD method, eight significant IMF modal components with different time scales and mean frequencies are generated to analyze the original signal here. Subsequently, the VMD decomposition results of turbulence shear in the time domain are extracted to their respective IMF modes, as shown in Figure 4 (left panels). Additionally, the features of the frequency spectrum for each extracted IMF mode in the frequency domain are transformed using FFT transform, as shown in Figure 4 (right panels). All the IMF modal components have different frequency components and significant differences in frequency bands. The VMD decomposition results of turbulent shear are effective to extract the multiscale and multi-frequency IMF components due to the contribution of the active turbulent eddies.

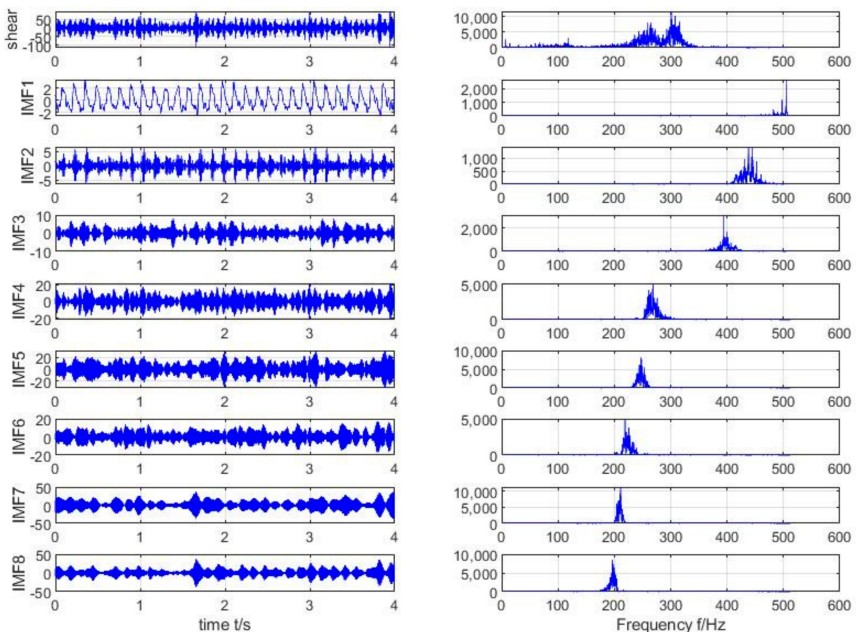

**Figure 4.** VMD decomposition results of shear signals in time-frequency domain.

Shear spectra in the wavenumber domain are commonly used to evaluate the order of magnitude of dissipation rates at which energy flows are irreversibly converted to heat by molecular friction from the largest scales to the smallest scales in the integrated dissipation process. Additionally, the Nasmyth spectrum, as a classical standard spectrum, is used to evaluate the measured shear wavenumber spectra. As shown in Figure 5, the shear wavenumber spectra density and Nasmyth spectra calculated using Equation (6) are given for four profiles. The order of cleaned TKE dissipation rates is resolved as low

as $10^{-9}$ Wkg$^{-1}$. The raw shear spectra and the cleaned spectra using the coherent noise removal method [23] conform well with the standard Nasmyth spectrum both in shape and level, which further proved that the platform is reasonably stable in the deep sea and the measured shear data are quietly effective.

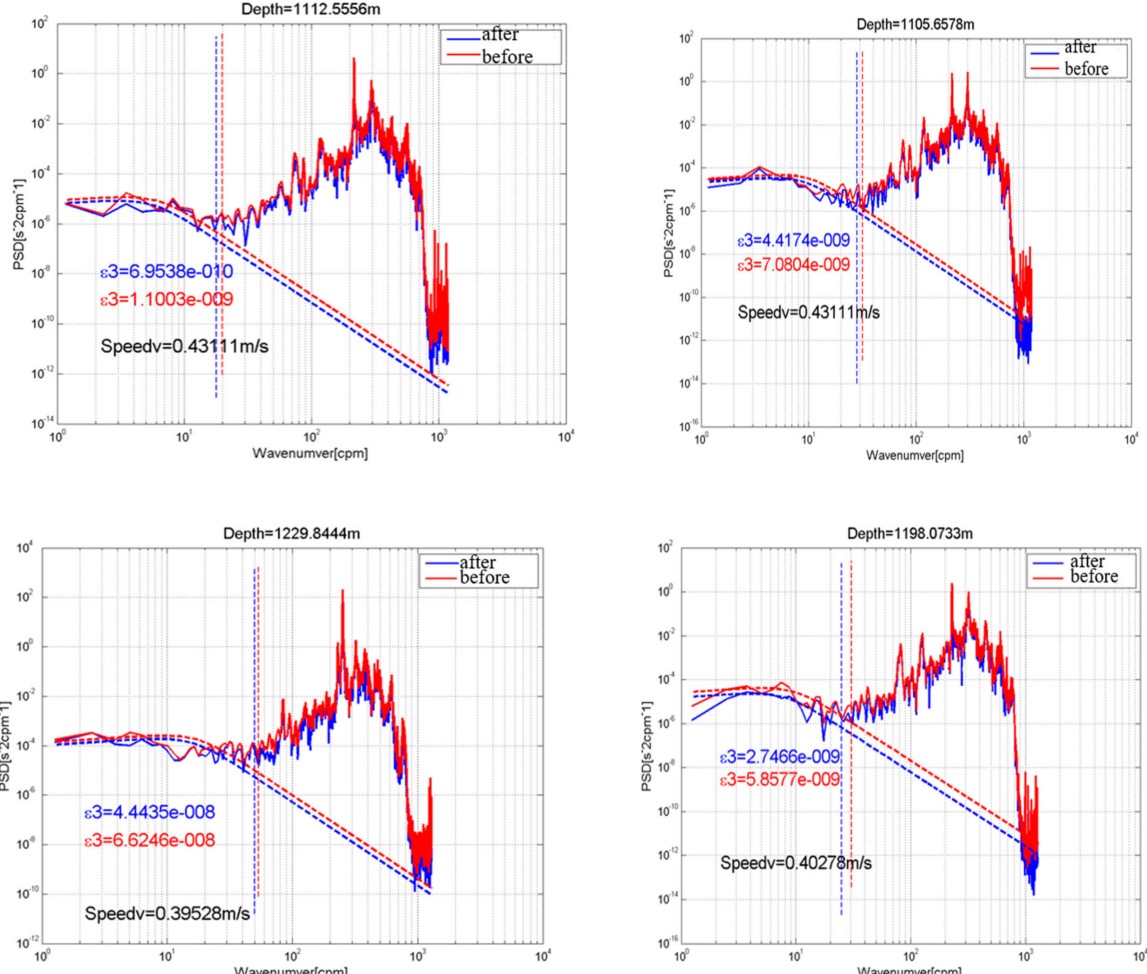

**Figure 5.** Shear power spectra density (PSD) in wavenumber domain. The red and blue lines denote the raw shear spectra and the cleaned spectra using the cross-spectrum method. The red and blue dashed curves denote the standard Nasmyth spectra, and the two vertical dotted lines denote the Kolmogorov cutoff wavenumber.

### 4.2. Temporal Features of Local Energy Cascade and Intermittency

To quantitatively measure the energy transfer process and exhibit the scale correlations, a number of multiscale IMF modes is estimated by means of an LIM approach. Local energy peaks and absolute values of phase difference of 8 (K = 8) neighboring IMF components are shown in Figure 6, where the top panels denote local intermittency measure peaks at two adjacent time scales, and the bottom panels refer to the absolute values of phase difference at two neighboring modal components. Through computing the corresponding average time scale of IMFs using a wavelet transform method, it is shown that the time scales are increasing with the number of the IMF modes (Figure 7), and the number of IMF modes has no effects on analyzing multiscale properties in the energy transfer process. By comparing the overlapped LIM energy peaks and corresponding phase differences of all the IMF modes, quantitative correlations of multiscale modes and properties of turbulent cascade and local intermittency are identified. The overlapped local energy peaks and the occurrence of a simultaneous phase between adjacent scales indicate an energy transfer between two adjacent time scales. When the LIM energy peaks of the intermittent

events in the dissipated process overlap, the phase difference between two neighboring scales can be nearly negligible; this phenomenon suggests that turbulent energy is being transferred between two neighboring scales, which have been highlighted in blue dotted lines (Figure 6). It is worth noting that the LIM peaks in the whole turbulent energy evolution process are discontinuous, indicating that this energy transfer process in a dissipated range occurs intermittently. Take IMF2 and IMF3 as an example. The absolute value of the phase difference between IMF2 and IMF3 can be negligible when their LIM energy peaks overlap twice between two neighboring scales at 0.54 s and 1.16 s (Figure 6b). It is illustrated that the overlapped energy peaks and the phases between IMF2 and IMF3 occur synchronously. For the last panel (Figure 6g), there are no simultaneous LIM peaks between two nonadjacent larger scales at 45 s and 180 s, indicating that the process of local energy cascade does not occur at two larger adjacent scales. In other words, the intermittent energy cascade can be easily found in smaller scales than in larger scales. In conclusion, phase synchronization is obvious at two neighboring smaller scales, and the simultaneous occurrence of cascade structures at neighboring pairs of smaller time scales is an effective indication of the local energy transfer process.

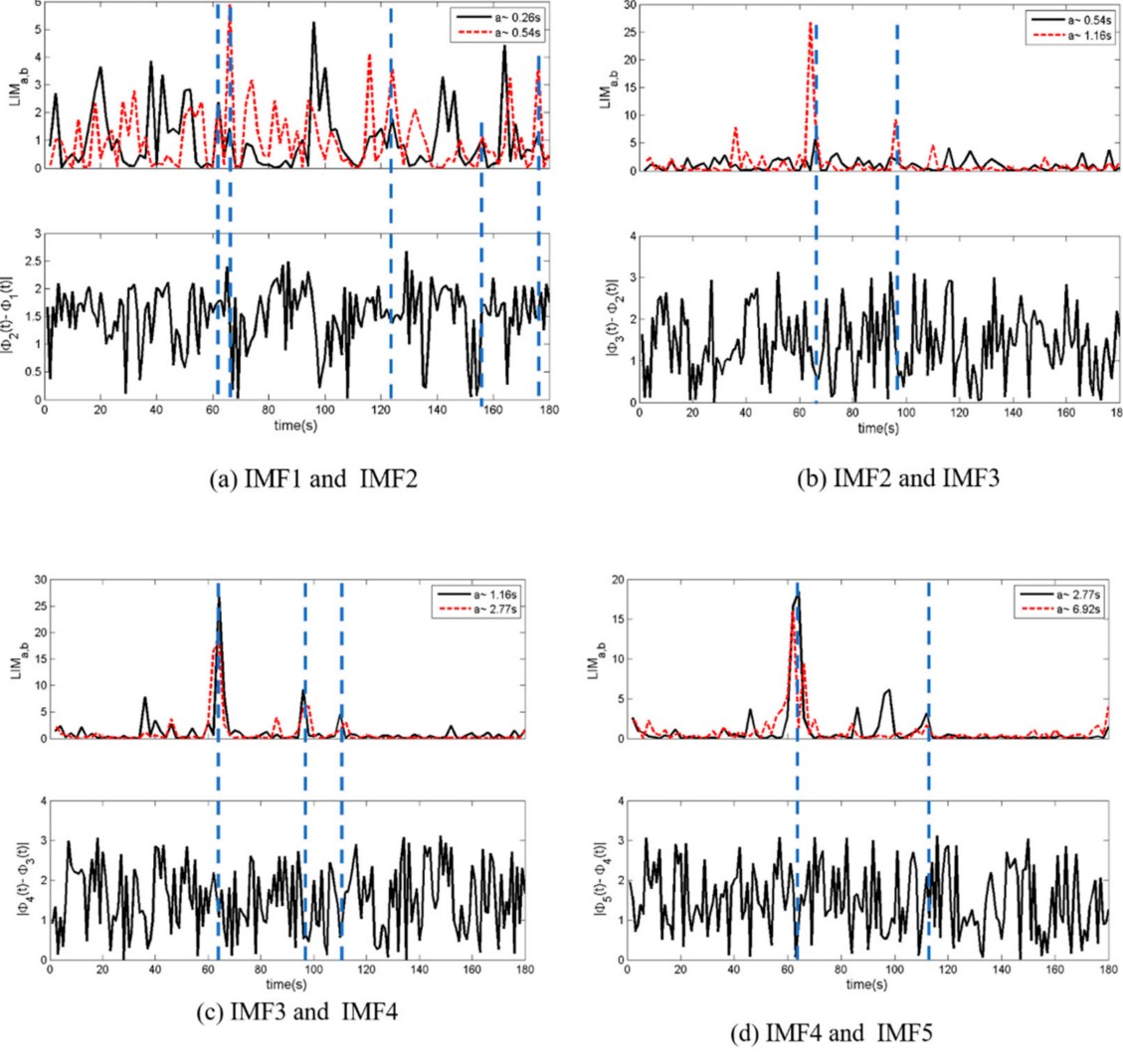

**Figure 6.** *Cont.*

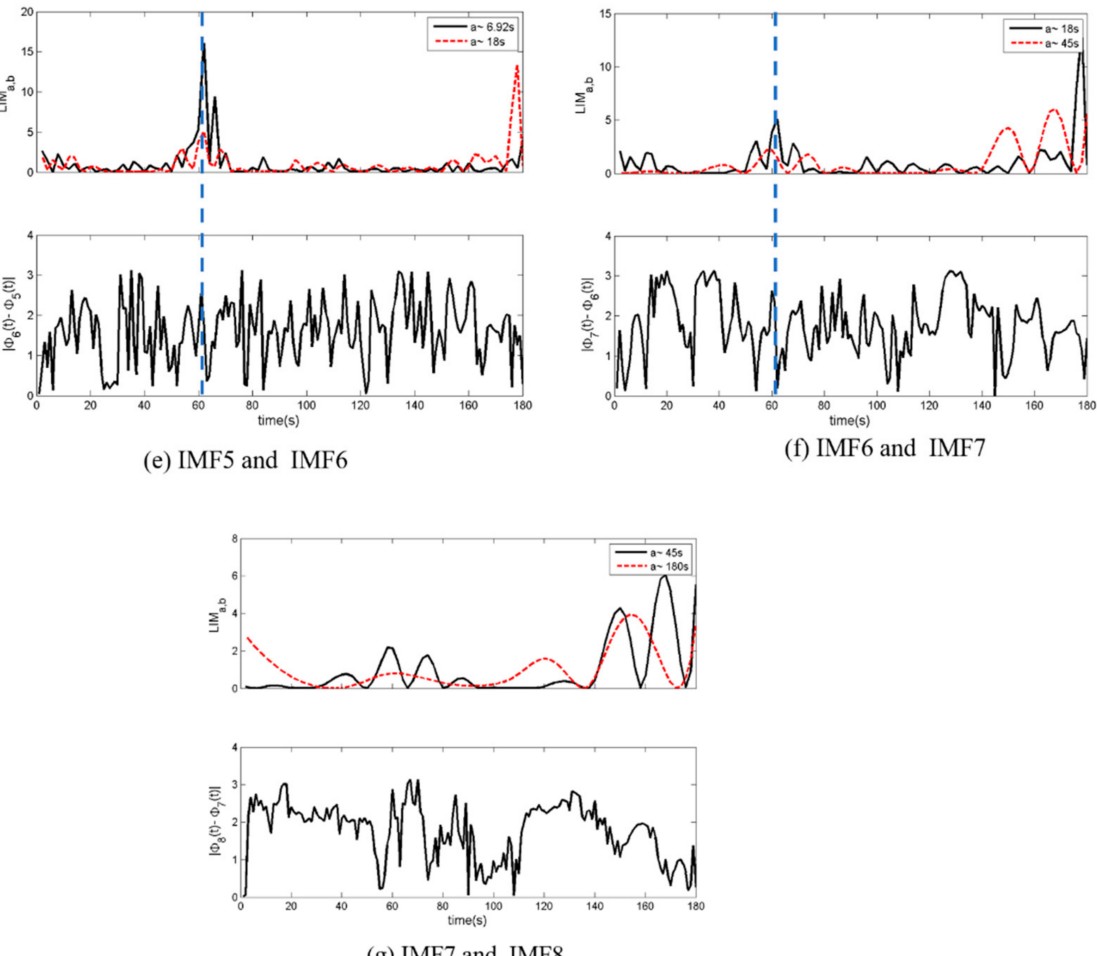

(e) IMF5 and IMF6

(f) IMF6 and IMF7

(g) IMF7 and IMF8

**Figure 6.** Phase correlations between IMF1 and IMF8 for profile1. (**a–g**) Top panels represent the local energy peaks at two neighboring scales; bottom panels represent the phase difference at two neighboring IMFs. The black line and red line respectively represent the LIM energy peaks of IMFi and IMF i + 1, where *i* = 1, 2, 3 . . . ., K, K is the number of modal components in VMD decomposition. The blue dotted lines denote that the phase difference is zero when LIM peaks are overlapping.

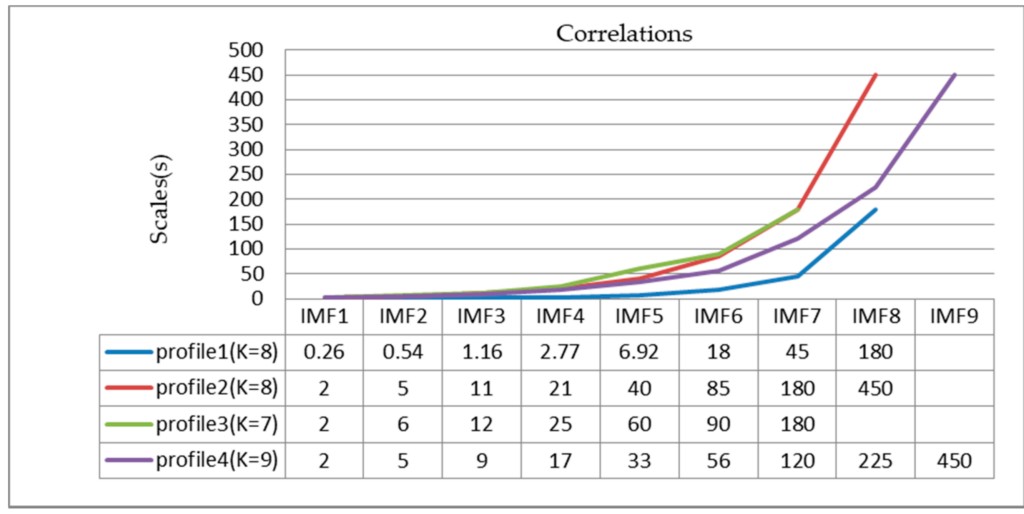

**Figure 7.** The correlations between the time scales and the number of modes K for four profiles.

### 4.3. Spatial Statistical Characteristics of Dissipation Rates

The dissipation rate of TKE ($\varepsilon$) is used to describe the energy dissipation in the turbulence mixing and dissipation process. The dissipation rates are denoted as the rate at which turbulent energy flow is irreversibly converted to heat from the largest scales to the smallest scales due to the interactions of molecular friction, which are also proportional to the shear variance that characterizes the intensity of turbulent shear [15]. The vertical distribution of dissipation rates between 600 and 1500 m depth is given in Figure 8. It is noted that the magnitude of the processed dissipation rates using the coherent cross-spectrum method [24] is as low as O $(10^{-9})$ Wkg$^{-1}$. The changes of the order of dissipation rates firstly increase and then decrease with the depth, and the variation trend of the vertical dissipation rates of the four profiles is particularly consistent, which further verifies the stable performance of the instrument.

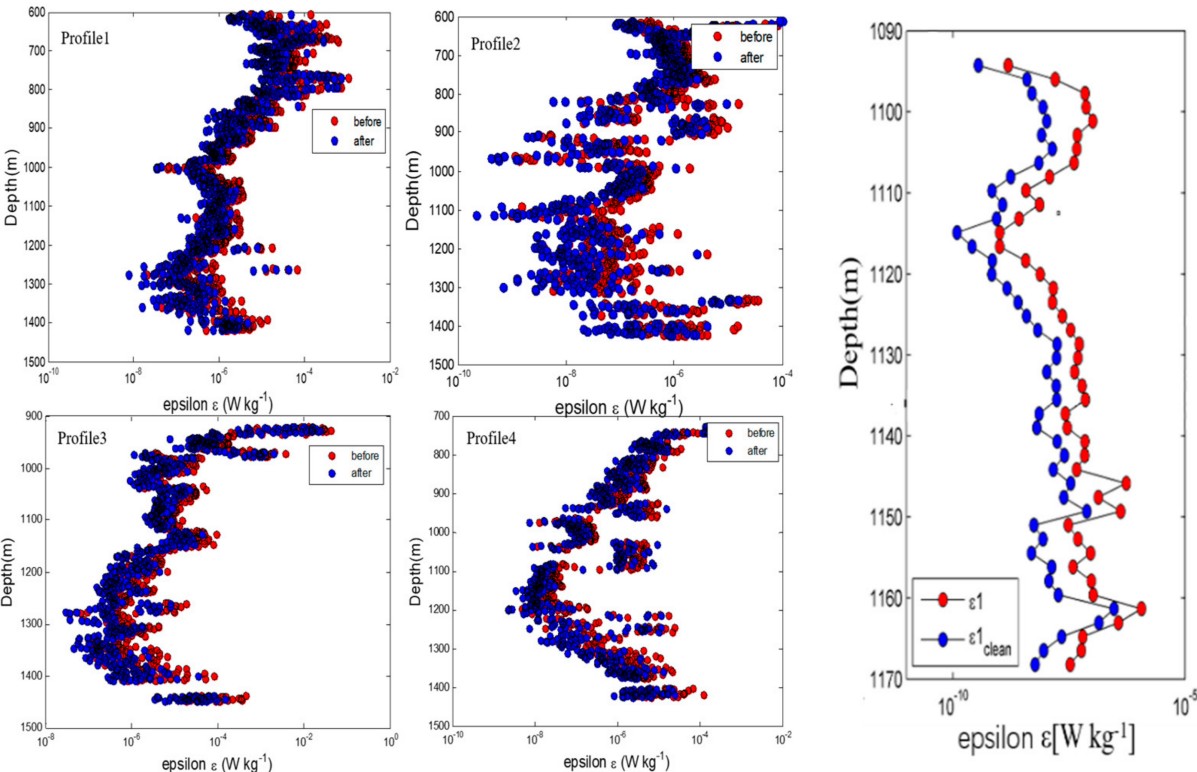

**Figure 8.** The raw dissipation rates (red dotes) and the cleaned dissipation rates (blue dotes) for four profiles using the classical cross-spectrum method for four profiles between 600 and 1500 m in deep sea. The right panel is the magnification of the dissipation rates between 1090 and 1170 m.

To further study the characteristics of vertical dissipation rates, a statistical cumulative probability distribution is proposed in Equation (7). The properties of energy cascade and the local intermittency of the turbulent flows can be identified by the flatness factor of the statistical normal distribution of the vertical dissipated energy. As shown in Figure 9, the dissipation of kinetic turbulent energy is approximated well by a standard normal distribution. As shown in Figure 10, the smaller the magnitude and the scale of dissipation rates, the larger the deviation of PDFs from a lognormality. With the increase in the dissipated scales, the distribution is closer to the normal distribution and the weaker the intermittency in the energy dissipation process. The present results suggest that the statistics of dissipation obey an important approximate lognormality in the dynamics process, and a few locations with high energy dissipation dominate the dominant position in the integrated energy dissipation process.

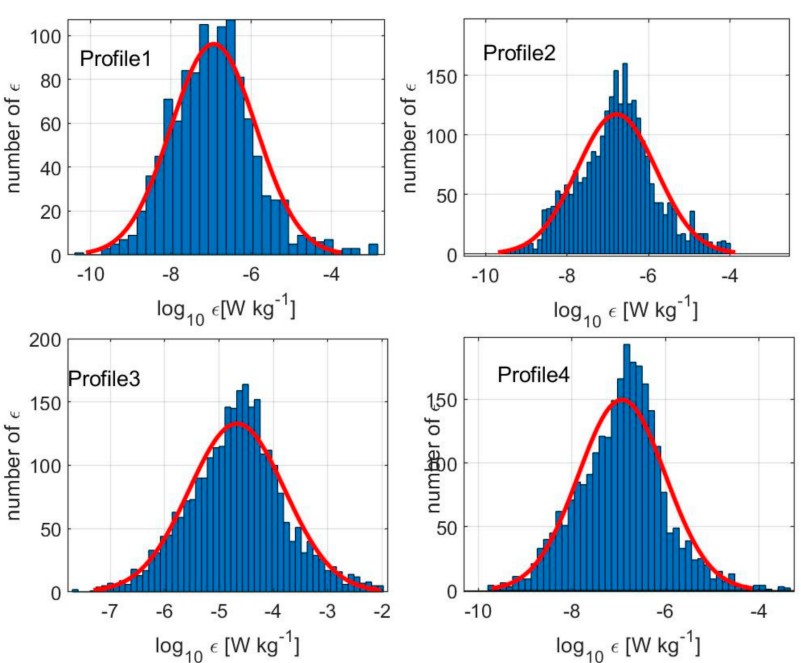

**Figure 9.** Probability distribution of dissipation rates for four profiles. The *x*-axis is the dissipation rates ($\varepsilon$) of turbulence energy and the *y*-axis is the number of $\varepsilon$ distributed at each interval. The red line shows the fitting cures for dissipation rates.

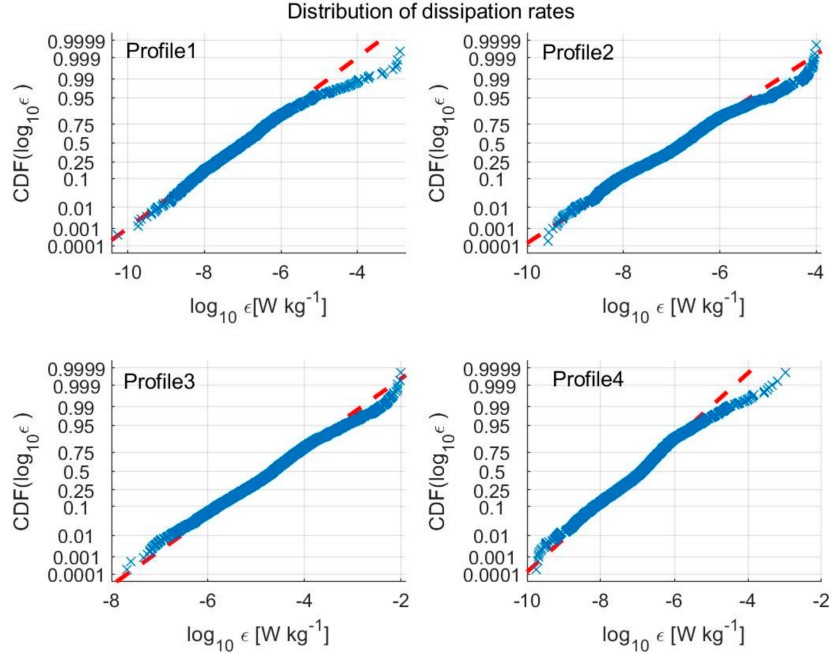

**Figure 10.** Cumulative probability distribution functions of dissipation rates for four profiles. The *x*-axis is the dissipation rates ($\varepsilon$) of turbulence energy and the *y*-axis is the probability, which ranges from 0 to 1.

## 5. Conclusions

In this article, a spatio-temporal synchronous analysis method associated with a non-recursive VMD decomposition, wavelet transform, and LIM method is proposed to quantitatively study the properties in the energy dissipation and cascade process. The main contributions of the proposed parallel extraction method of spatio-temporal features are summarized as follows.

(1) To capture the local temporal properties of turbulent intermittency and energy cascades in the ocean, an integrated quantitative method using a combination of the VMD, WT, HT, and LIM is proposed. The correlations between two adjacent IMF components are identified by the overlapped LIM energy peaks and the phase synchronization. Multiscale cascades and intensive intermittent events exist in the two neighboring time scales, which are characterized by phase synchronization in the energy evolution process. Meanwhile, the preliminary results find that the energy transfer phenomenon is not obvious between smaller or larger scales through comparing different time scales.

(2) Spatial statistical features of dissipation rates are synchronously studied to learn the spatial distribution features. The distribution of dissipations obeys an approximate normal distribution in energy dissipation area, indicating that they exhibit robust scales relations in measured turbulence data, and a few high dissipation areas dominate the whole dissipation process.

The proposed integrated method may have good prospects in improving our understanding of the turbulent energy transfer process and quantitatively identifying energy cascade features in the ocean dynamic systems. Meanwhile, the number of decomposition modes and the selection of the penalty parameter in the VMD method are necessary to be optimized further by some optimization algorithms, such as the gray wolf optimizer (GWO), particle swarm optimization (PSO), and the whale optimization algorithm (WOA) in the future work.

**Author Contributions:** Methodology, X.L. and D.S.; writing—original draft preparation, X.L. and X.W.; writing—review and editing, H.Y.; formal analysis, X.L. and X.W.; funding acquisition, X.L. and H.Y. supervision, Y.N. All authors have read and agreed to the published version of the manuscript.

**Funding:** The research work is supported by the National Natural Science Foundation of China, grant number 62001262, 61871354, 61727806 and the Natural Science Foundation of Shandong Province, grant number ZR2020QF008.

**Institutional Review Board Statement:** Not applicable.

**Informed Consent Statement:** Not applicable.

**Data Availability Statement:** Not applicable.

**Acknowledgments:** We would like to thank the Ocean University of China for the platform of the vessel "Dong Fang Hong 2"and thank the whole teams of the research and development for their help with this experiment. We also deeply thank the reviewers and the editor for their constructive criticism of an early version of this manuscript.

**Conflicts of Interest:** The authors declare no conflict of interest and the manuscript is approved by all authors for publication.

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
