# Peer review of "An Integrated Spatio-Temporal Features Analysis Approach for Ocean Turbulence Using an Autonomous Vertical Profiler"

_applsci, doi:10.3390/app11209455_

Round 1
Reviewer 1 Report
The paper is very interesting and contains valuable information on turbulent energy cascade in Ocean. The authors should highlight the novelty of the study and the gaps filled. It is not very obvious. What is the contribution to the state of the art? The readers should be able to see this in the conclusions section.
For the measurements, two orthogonal shear probes and a vibration sensor were used. The reviewer want to know about a calibration procedure and result, and an uncertainty of the results. It is good information for readers.
Author Response
Dear reviewer,
Thanks for your comments, and we have carefully revised the manuscript according your comments.
Please see the attachment.

Reviewer 2 Report
This is an interesting paper using an experimental method to investigate the ocean turbulence and the spatial-temporal features in energy transfer process. The paper is well organised and well written. The research methodology adopted in the paper is also appropriate and the research results support the conclusions. So I am happy to suggest to accept this paper for publication in the Journal Applied Sciences subject to improving the paper based on the following comments:
- Research question to be dealt with in the paper needs to be highlighted in Abstract to attract reader's interest;
- Figure 1 should be removed for two reasons. Usually, figure will not appear in Introduction section of a paper. Secondly, the figure cannot accurately describe the energy transfer process of ocean turbulence.
- Ocean turbulence can be affected by many factors, including the influence of the data measurement equipment itself. I am not going to question the data measurement method used in the paper. But in the paper, it is necessary to note which factor or factors that this paper mainly consider? Wave, tidal current, etc.?
- In Line 166, you mentioned 'improved VMD'. In line 361, you said 'adaptive VMD'. I was confused by relevant descriptions as you only considered VMD in your work.
- In Conclusion, you also said 'The correlations between two adjacent modes are molded by phase-synchronization and the localized peaks of energy'. What did you mean 'molded'? A very confusing word.
- The label of Y-axis in Figure 10 was missing. The unit of Probability in Figure 11 was missing either.
- From Figure 9, can you see any effect of signal filtering? I cannot see anything positive from the results.
- A proof reading is essential before re-submission. A few languages used in the paper have been identified that need to be improved, but there may be more. For example
a) You used 'abovementioned' for a few times. Should it be 'aforementioned' or 'above-mentioned'?
b) In line 56, you said 'Literature research shows'. Should it be 'Literature review has shown'?
c) In line 102, you said 'The outline of this paper is organized as follows'. This might be a wrong description. How 'outline' is organised?
d) In line 129, you mentioned two directions, horizontal and lateral. Is lateral also a horizontal direction?
e) In line 130, you mentioned a direction 'heading'. What is that direction?
f) In the paper (line 166-174), you said that VDM can conquer the problem of mode mixing that was observed from EMD. I cannot fully agree with this because mode mixing still exists in the results of VMD, see your own results shown in Figure 5.
g) In line 261, you said 'There is no interaction between different IMF components'. What do you mean herein?
Author Response
Dear reviewer,
Thanks for your comments, and we have carefully revised the manuscript according your comments.
Please see the attachment.
Thanks and best wishes,
Xiuyan

Round 2
Reviewer 2 Report
The authors have improved the paper significantly and addressed all concerns raised by the reviewer. Now, the reviewer is happy to suggest to accept this paper for publication in the Journal Applied Sciences.